# Effect of *Moringa Oleifera* fortified porridge consumption on protein and vitamin A status of children with cerebral palsy in Nairobi, Kenya: A randomized controlled trial

**Janet Kajuju Malla**[1,2]*, **Sophie Ochola**[1], **Irene Ogada**[1,3], **Ann Munyaka**[1]

**1** Department of Food, Nutrition and Dietetics, Kenyatta University, Nairobi, Kenya, **2** Department of Human Nutrition and Dietetics, Technical University of Kenya, Nairobi, Kenya, **3** Department of Applied Human Nutrition, Mount Saint Vincent University, Halifax, Canada

* janetmalla66@gmail.com

## Abstract

### Background

Malnutrition due to inadequate dietary intake is commonly reported in children with Cerebral palsy (CP). Poor dietary intakes are majorly caused by feeding dysfunctions secondary to oro-motor impairment characteristic of the condition. Strategies that improve nutrient densities in foods can help enhance nutrient intakes by these children.

### Objective

This study investigated the effect of consumption of fermented finger millet porridge fortified with *Moringa oleifera* leaf powder (MoLP) on the protein and vitamin A status of children with CP.

### Design

A randomized controlled trial was conducted among 113 children aged 5–11 years with CP. The study had two arms (intervention [N = 57] and control [N = 56]). The intervention group received a daily serving of fortified finger millet porridge for 3 months while the control group received non-fortified finger millet porridge servings. All children received the same amounts of porridge servings. The levels of serum albumin and retinol between the groups were compared at both baseline and end line. The BMI-for-age Z-scores (BMIAZ) and morbidity prevalence of the children were also assessed.

### Results

At baseline, the two study groups were similar in all demographic and socio-economic characteristics, nutrient intakes, serum levels of albumin and retinol, weight status and morbidity. At end line, the children from the intervention group had significantly higher intakes of vitamin A at 717.12±432.7 µg/d (p = 0.038) and protein at 44.367±17.2 g/d (p = 0.031) respectively. The serum nutrients levels increased significantly from baseline by 0.456±0.12 g/dL

**Data Availability Statement:** All relevant data are within the paper and its supporting information files.

**Funding:** This was a grant - No. NRF/122/2016. Funded by the Kenya National Research Fund (NRF). URL: researchfund.go.ke The grant was awarded to Janet Malla and it only catered for field work and analysis of data. The funders did not play any role in study design, data collection & analysis, decision to publish nor preparation of the manuscript.

**Competing interests:** The authors have declared that no competing interests exist.

(p<0.001) for albumin and by 0.243±0.10 µmol/L (p<0.001) for retinol among children in the intervention group. Among the children in the control group, the changes in the levels of both serum albumin 0.012±0.07 g/dL (p = 0.868) and serum retinol [0.0021±0.02 µmol/L (p = 0.890)] were not significant. At endline, the BMI-for-age Z-scores results showed that 10.52% and 34.0% of children from intervention and control group respectively were under-nourished [χ2 = 30.985; p = 0.037]. Among the children in the intervention, group there was a significant change in the weight status between baseline and endline (p = 0.036). The weight status among children in the control group was not significantly different between baseline and endline (p = 0.109). Significant difference in morbidity prevalence between the two groups was also observed at endline (p = 0.003) with the prevalence being 24.6% and 51.8% among children in the intervention and control group respectively.

## Conclusion

Consumption of *M. oleifera* fortified porridge significantly improved the children's serum albumin and retinol levels, as well as BMIAZ.

## Registration number and name of trial registry

The trial is registered at Pan African Clinical Trials Registry, number PACTR202107669905145 URL link: https://pactr.samrc.ac.za/.

## Introduction

Cerebral palsy (CP) is a group of neurologic disorders typically caused by a non-progressive lesion or abnormality of the developing brain that appears in infancy or early childhood and permanently affects body movement, muscle coordination, and balance [1]. It is the most common cause of physical disability in childhood and constitutes a significant health problem with major effects throughout the lifespan [1–4]. The global incidence of CP is approximately 2 to 2.5/1000 live births, with prevalence varying widely from country to country [5,6]. Despite concerns of underreporting, this prevalence has been suggested to be even higher in developing countries throughout Africa [7] and is estimated at up to 10 cases per 1000 births[5,8]. Higher prevalence in Africa is likely due to the level of perinatal complications such as birth asphyxia and neonatal infections [5]. There is limited data on the current prevalence rate of CP in Kenya.

A number of studies have reported increased prevalence of morbidity and mortality secondary to compromised nutritional status among children with CP compared to their normal counterparts in the same age group [3,9–11]. According to the UNICEF conceptual framework on the determinants of malnutrition [12], dietary intake is one of the immediate determinants of nutritional status. Children with CP commonly suffer nutrient deficiencies as a result of compromised dietary intakes due to feeding dysfunctions such as oropharyngeal dysphagia associated with the condition [13]. Strategies to enhance nutrient densities of foods consumed by children with CP are thus critical to ensuring that they meet their requirements of essential nutrients for optimal growth and health [10,14,15]. Among the common nutritional concerns in children with CP are Protein-Energy Malnutrition (PEM) and vitamin A deficiencies (VAD) [16,17].

The potential of *M. oleifera* for use as a fortificant in different food formulations including soups and complementary foods has become a subject of research interest due to its high

nutritional value which is widely acknowledged in literature [18–20]. Improvements in protein malnutrition, VAD and calcium and iron deficiencies in children have been reported after interventions with *M. oleifera* [21]. In a related publication, the authors have reported the potential of the *M. oleifera* fortificant to improve the protein and vitamin A content of fermented finger millet porridge [22].

The Food and Agricultural Organization is currently promoting the use of indigenous and/ or underutilized crops readily found in various localities as an economically sustainable strategy to enhancing food security and addressing common deficiencies in local populations [23,24]. There is, however, a scarcity of studies addressing the nutritional viability of local/ indigenous crops to treat malnutrition in disease contexts particularly in neurological disorders such a cerebral palsy. The lack of rigorous and systematic tests of the nutritional efficacy of local food fortificants to mitigate malnutrition in disease context, constituted a gap this study aimed to address. The study was undertaken to investigate the effect of consumption of *M oleifera* fortified porridge on protein and vitamin A status in children with cerebral palsy.

## Methods

### Study site

The study was conducted at Little Rock Day Care Centre located in Kibra informal settlement in Nairobi County from May to July 2018. The Little Rock Day Care Centre center serves as a rehabilitation center for children with special needs. Kibra is one of the most densely populated informal settlements in Kenya and with the highest poverty rates with over 70% of the population living below the poverty line. Sanitation and hygiene conditions are very poor and physical infrastructure including housing and road access are at their poorest condition.

### Study design

This was a randomized controlled trial (RCT) investigating the effect of the consumption of *Moringa oleifera*-fortified finger millet porridge on protein and vitamin A status of children with cerebral palsy in Nairobi Kenya. The study comprised of two study arms: the control group (receiving fermented finger millet porridge) and the intervention group (receiving fermented finger millet porridge fortified with *M. oleifera* leaf powder). The study was approved by the Kenyatta University Ethical Review Committee (KUERC), Ref. No. KU/ERC/ APPROVAL/VOL.1 (83). Written informed consents were obtained from caregivers of the children who were the respondents.

The trial whose results are being published was registered retrospectively. This was mainly because of limited prior knowledge on trial registration. This however, did not in any way influence the reporting of the findings and the research methodologies. The authors confirm that all ongoing and related trials for this intervention are registered.

### Sample size determination

The study adopted the formula by Noordzij [25] for sample size calculation in RCT studies. Calculation of the sample size was based on the main primary outcome (serum albumin levels); 80% power of the test and a 5% level of significance ($\alpha = 0.05$) according to the recommendations by Cohen, [26] and Noordzij et al., [26]. Both the population means and variance were derived from the results of the pilot study conducted for this study. The calculated sample size was **52** participants (caregiver-child pairs) per group. This sample size was inflated by 10% to cater for attrition, hence the final sample size was; 52+5 = **57** participants per group totaling = **114** caregiver-child pairs for the two study groups.

### Randomization

Upon recruitment, the children were randomly allocated on a 1:1 ratio to the two study groups (control and intervention) by the researchers. Randomization was conducted by an independent biostatistician using a computerized random number function in Microsoft Office Excel 2008. The numbers generated were printed on a piece of paper, uniformly folded, and the caregivers were asked to pick one. The children were placed into the groups depending on the number which the caregiver picked and which had been allocated to by the randomization process.

**Recruitment process and trial profile of the study subjects.** A total of 114 children were recruited and randomized into the study. All those from the intervention group completed the full course of treatment and follow-up for the three-month duration of the study (May-July 2018). One child from the control group was lost to follow up because of complications developed during the study though not related to the study treatment ([Fig 1]). Both the experimental porridge (*Moringa oleifera* fortified fermented finger porridge) and the control porridge (non-fortified fermented finger millet porridge) were well tolerated. No adverse reactions attributable to the products occurred during the study period.

### Study participants and screening

The study targeted children ages 5–11 years diagnosed with cerebral palsy by a medical doctor, attending Little Rock Daycare Center in Kibra Informal settlements and their caregivers. Children with cerebral palsy who had chronic ailments that would independently affect food intake such as; cardiovascular diseases, cancer, renal dysfunction, and liver problems were excluded. The children who met the inclusion criteria and whose caregivers were willing to participate in the study by giving their informed consent, were randomly assigned to the intervention or control groups by the researchers.

### Selection and training of research assistants

**Selection and training of the cooks.** Two cooks with diploma level of education in food and beverage service were recruited and trained on the standard recipe of the porridge and the serving portions. They were trained on their roles in preparing the porridge and weighing out the servings as well as weighing and recording the left-over portions from each of the study participants- the children. The cooks worked closely with the data collectors.

**Selection and training of data collectors.** Four data collectors with a minimum of diploma in Food, Nutrition and Dietetics were recruited into the research team. Prior experience in surveys was an added advantage. The data collectors were trained for a period of 2 days by the principal researcher on: their roles and expectations; research ethics; recruitment of the study participants; how to administer the data collections tools and the entire data collection procedures.

### Blinding design

The biostatistician, the cooks and the data collectors, study participants, caregivers and the workforce at the children's centre were all blinded to the study hypotheses. It was only the researchers who were aware of the study hypotheses. The cooks were instructed not to disclose the recipe of the coded porridge samples fed to the children to any one including the data collectors and the participants.

### Description of interventions

**Control group (CG).** Children in the control group received porridge prepared from fermented finger millet flour. The porridge was prepared at the daycare center by the caterer and

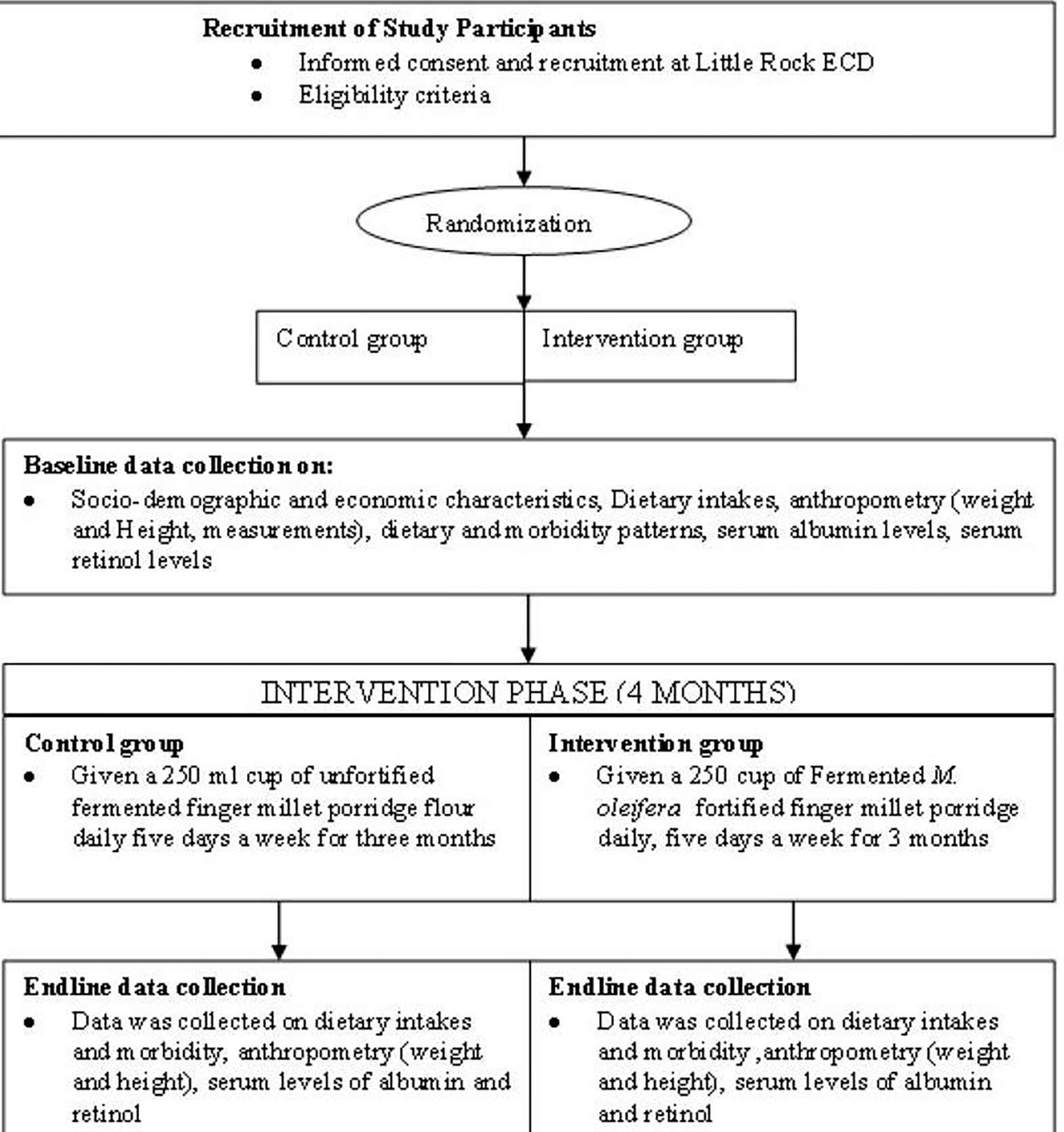

**Fig 1. Trial profile of the study participants.**

cooks, following a standard procedure, explained in a related publication [22], and fed to the children by data collectors, both under the supervision of the researchers. Each child received a daily portion of porridge equivalent to 250ml in a cup at 10 am, five days a week, for three

months. The left-over porridge by the children was measured and the amount of porridge consumed by each child determined daily for the entire study period (Fig 2).

**Intervention group.** Children in the intervention group received porridge prepared from fermented finger millet flour fortified with 10% *M. oleifera* leaf powder. As presented in a related publication, 10% fortification rate was found to significantly improve the vitamin A and protein content of the porridge without significantly compromising its sensory acceptability, that is colour, texture, taste and overall acceptability [22]. The same procedure used for control group was followed in the preparation and administration of the fortified porridge for the children in the intervention group. (Fig 2). Pictures of samples of the prepared porridges is presented in S1 Fig.

## Study outcomes

The primary study outcomes in this trial were the improved protein and vitamin A status of the children. Protein status was determined by the levels of serum albumin which is the most abundant protein in the blood and also the major carrier of free fatty acids in the blood, whereas vitamin A status was indicated by serum retinol levels. The secondary outcomes were changes in weight status measured by BMI-for-Age Z-scores (BMIAZ) and morbidity status assessed by the prevalence of illness based on a 2-week recall.

## Data collection tools

The study questionnaires were developed and face-validated through multiple pre-tests, and finally pilot- tested and refined for clarity and accuracy. Two questionnaires were used to collect data; a baseline questionnaire and a questionnaire used at the end of the study (at three months). The questionnaires solicited information on the 24-hour dietary recall, anthropometry (weight and height measurements), morbidity prevalence based on a 24-hour recall. The questionnaire had a biochemical test schedule to record levels of serum albumin and serum retinol. In addition, the baseline questionnaire solicited information on the socio-demographic and economic status of the study participants.

## Pilot study

A pilot study was conducted on a sample of 14 caregiver-child pairs (about 11% of the sample size for the main study sample) from a Centre in Nairobi comparable to the study site. Piloting was conducted to validate and standardize the study procedures and research instruments. The pilot study also accorded the research assistants the opportunity to practically apply the skills learnt during the training, and experience the possible challenges that would be faced during the main study. The feedback was used to make appropriate and necessary changes in the study procedure and data collection tools before commencement of the study. The findings of the pilot study are presented in S2 Table.

## Collection of data at baseline and end line

Except for the data on the demographic and economic characteristics of the study respondents which was collected at only baseline, the rest of the data was collected at both baseline and endline. Information on the demographic and socio-economic profile of the study subjects was collected at baseline to establish homogeneity or differences in the two study groups that could have a bearing on the study outcome variables.

Data was collected using the same methods on the following variables at both baseline and endline.

**Collection of blood samples for determination of protein and vitamin A status of the children.** A certified phlebotomist collected 5.0 mL of venous blood by venipuncture of a

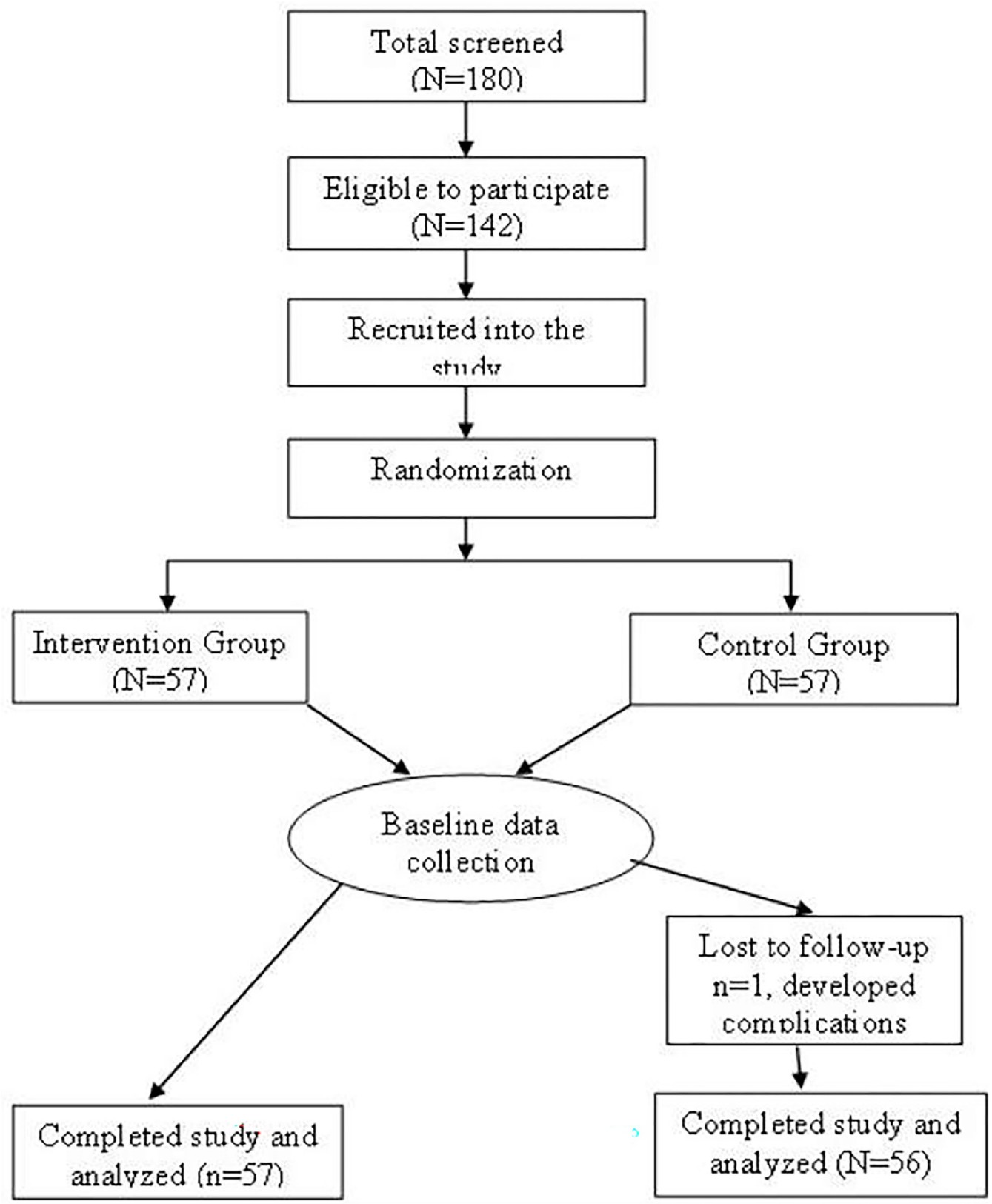

**Fig 2. Schematic presentation of the research design.**

peripheral vein using disposable syringes. The specimens were collected in trace-element-free tubes, immediately wrapped in aluminum foil, to shield them from light, then put in portable freezers for transportation to the laboratory for biochemical analysis where they were stored at 4°C until centrifugation.

**Anthropometry measurements.** Anthropometric measures were taken using standardized protocols [27]. Weight was measured in kilograms using a zero-calibrated digital weighing scale (Seca Robust 813) and recorded to the nearest 0.1 kg, following standard guidelines (i.e., removing extra clothing, standing still). Alternatively, for children not able to stand alone, the weight was calculated as the difference between the caregiver's weight with and without the child. Standing Height was measured using a stadiometer (Seca 217) and recorded to the nearest 0.1 cm. For children with deformities, the knee height (the height from the knee to the heel) was taken and used to estimate the overall height using the equation* height = (2.69 × knee height) + 24.2 [28]. Knee height was recorded for all children to the nearest millimeter. Following World Health Organization (WHO) best-practice procedures, height, length, and knee height measurements were obtained twice by two independent observers: and the average taken as the final measurement if they agreed to within less than 5mm, or else both re-measured until agreement achieved [29].The BMI of the children enrolled in the study was calculated from weight and height measurements as weight (in kg)/height$^2$ (in m) and BMIAZ determined using WHO 2006 BMI-for-age reference tables for children and adolescents (5–19 years). Z-scores were then used to categorize the children according to their weight status.

## Dietary intakes

Non-consecutive two-day 24-hour dietary recalls were conducted to determine the nutrient intake per day and hence the adequacy of the nutrients. The two recalls were conducted within a two-week interval with one being conducted on a weekday and the other on a weekend to minimize random error and to capture the variation between weekdays and weekend days. The 24-hour recall was undertaken by trained research assistants using a standard protocol. The data collected included a description of the foods eaten; the cooking methods; brand names (e.g., for cereals, processed snacks); and the quantity of food consumed which was estimated using the common household measures. Each food item consumed was linked to the equivalent food in the Kenya Food Composition Tables for determination of nutrient content per unit (100g) and consequent calculation of nutrient intakes. The average nutrient intakes were calculated from the two recalls. The multiple pass approach consisting of five steps was adopted in conducting the 24-hour dietary recall interviews since it yields the most accurate data and best recall [30]. The steps included (1) Quick list, to collect a list of foods and beverages consumed during the previous day (i.e., over past 24 hours) (2) Forgotten foods, to probe for foods forgotten during the quick list (3) Time and occasion, to record time and eating occasion for each food (4) Detail cycle, to record detailed description, amount, use of condiments, cooking methods, brands of foods if purchased, and other information as review of 24-hour day and (5) Final probe, to probe anything else consumed.

## Morbidity

Morbidity data were collected, based on a 2-week recall period to determine the prevalence (if a child had suffered illness), frequency (number of times a child fell ill), and types of illnesses reported.

## Data analysis

**Biochemical analysis for nutrient intake.** Retinol was extracted from serum with ethanol containing butylated-hydroxytoluene (BHT). Retinol was determined by HPLC (Thermo

Scientific Accella LC system; Thermo Fisher Scientific) with spectrofluorimetric detection (λex 340 nm; λcm 460 nm), but serum albumin was quantified using bromocresol green (BCG) ASSAY method. The absorbances (Abs) of standards and the samples was read in a UV-Vis spectrophotometer (Beckman DU-650, Fullerton) at 545 nm against the blank and albumin concentration in the samples was determined according to the general formula:

*Albumin (g/dl) = (Abs sample/ Abs standard) X Concentration of standard.*

### Anthropometric data analysis

WHO Child Growth Standards (2006) [31] were used to interpret the nutritional status of the children because this is what is currently used in Kenya: Cut-offs for BMIAZ were set at ≤-2SD for wasting/thinness, >+1 for overweight and >+3 for obesity.

### Dietary intake data analysis

The 24-hour recall data was entered into the Nutri-survey software for processing using the Kenya Food Composition Table as the reference for nutrient composition. In instances where a food item could not be found in the Kenya Food Composition Table, it was added to the database using food manufacturing websites and other Food Composition Tables. Nutrient data obtained from the analysis were compared with the Recommended Daily Allowances by UNICEF/WHO (2002) to determine the adequacy of nutrient intakes.

Statistical analyses were performed using Statistical Package for Social Sciences (SPSS) version 19.0 software. All statistical tests were two-tailed with an alpha value of 0.05. Summary statistics (means, percentages and SDs) were used to describe all variables of interest. Data were tested for normality to decide between parametric and non-parametric tests. Parametric tests including t-test, and Pearson product-moment correlation were used to analyze normally distributed data while non-parametric alternatives such McNemar-s test were used for non-normally distributed data. The chi-square test, Marginal homogeneity test, and McNemar's test were used to test differences in the two study groups for categorical variables such as education, marital status, occupation and BMIAZ. Independent t-test was used to test for differences between the intervention and the control groups for continuous variables such as age, nutrient intakes, serum levels of albumin and retinol. The paired t-test was used to compare differences in continuous variables (nutrient intake, serum albumin and retinol levels) between baseline and end line for each study group and to determine the magnitude of change in nutrient intakes and serum levels. The wealth index was computed by means principal component analysis (PCA) using data on household asset ownership (Fig 3). Included in the analysis were key items including land, car, motorcycle, mobile phone and bank account ownership. The PCA revealed two components that explained 63.64% of variance. Component one eigenvalue = 1.42 (variance explained = 35.59%), component two eigenvalue = 1.12(variance explained = 28.05%). All the measures loaded onto one of the two components. Component one was made of bank account, bicycle and car. The second factor was made of land, and bank account. Component one was thus taken as the wealth index. The wealth index was recoded into a categorical variable by dividing the household into 5 groups.

## Results

### Comparison of study groups at baseline

**Socio-demographic and economic baseline characteristics by study groups.** No differences were found in socio-demographic and economic characteristics of the participants of the two study groups at baseline, an indication that randomization was successful (Table 1).

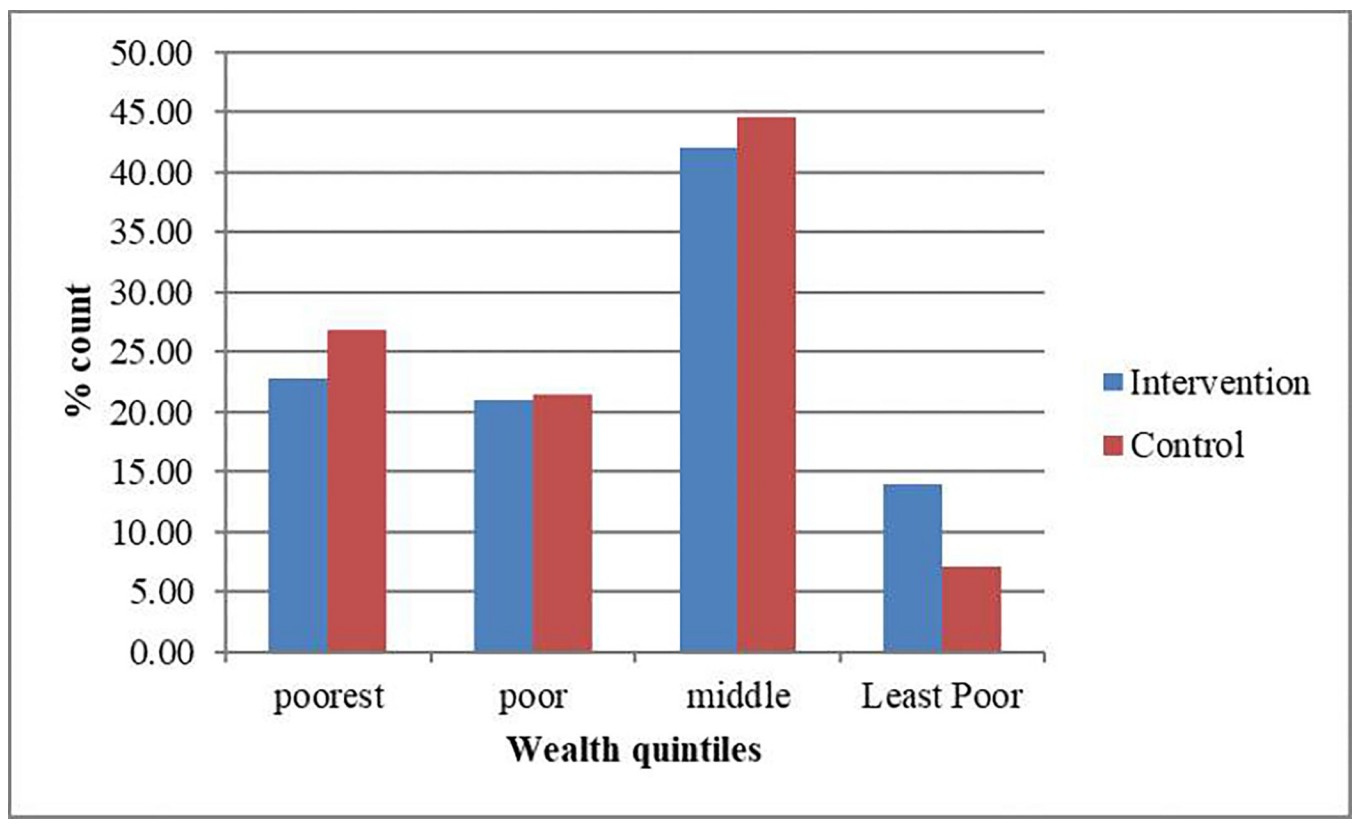

**Fig 3. Wealth quintile distribution between the study groups.**

**Dietary intakes by study groups at baseline.**   The mean daily intakes by the intervention and control groups were 30.68±14.77g and 30.50±16.93g respectively for proteins and 456.12 ±374.14μg and 477.45±373.5μg respectively for vitamin A. There were no significant differences in the intakes of nutrients between the study groups at baseline (Table 2).

**Levels of serum albumin and serum retinol at baseline by study groups.**   The baseline serum albumin levels were 3.067±1.09 and 3.186±1.16 g/dL for the intervention and control group respectively (Table 3). There was no significant difference in baseline serum albumin levels between control and intervention group children (t = -1.088; p = 0.280). The serum retinol levels at baseline were at 0.471± 0.05 and 0.468 ± 0.02 μmol/L in the intervention and control group respectively. There was no significant difference in baseline serum retinol levels between the children in the control and intervention group (t = -0.042; p < 0.966) (Table 3).

**Weight status of the study participants at baseline by study groups.**   The BMI of the study participants were calculated from anthropometric measurements and BMI-for-age Z-scores determined using WHO 2006 BMI-for-age Reference for Children and Adolescents (5– 19 years). The Z-scores were then used to categorize the children according to their weight status (Table 4). At baseline, 38.6% and 33.9% of children from intervention and control groups respectively were underweight whereas 52.6% and 55.4% of children from intervention and control groups respectively were in the normal weight range (between ≥-2SD and +1SD). A small proportion of the study children, i.e. 5.3% and 7.1% of children from intervention and control groups respectively, were overweight while 5.5% and 3.6% of children from the intervention and control groups respectively, were obese. No significant differences existed in the weight status between the two study groups (t-test, p = 0.856) at baseline.

**Table 1. Socio-demographic characteristics of caregivers.**

| | Intervention, N = 57 | Control, N = 56 | Total N = 113 | p-value |
|---|---|---|---|---|
| **Caregiver characteristics** | | | | |
| Age mean (SD) | 35.8(6.5) | 37.8(6.5) | | 0.103[a] |
| **Marital status** | **n (%)** | **n (%)** | **n (%)** | |
| Married | 26(45.6) | 21(37.5) | 47(41.6) | 0.315[b] |
| Single | 14(24.6) | 16(28.6) | 30(26.5) | |
| divorced | 6(10.5) | 12(21.4) | 18(15.9) | |
| cohabiting | 11(19.3) | 7(12.5) | 18(15.9) | |
| total | 57(100.0) | 56(100) | 113(100) | |
| **Education** | | | | |
| No formal | 2(3.5) | 1(1.8) | 3(2.7) | 0.442[b] |
| primary | 21(36.8) | 24(42.9) | 45(39.8) | |
| secondary | 14(24.6) | 19(33.9) | 33(29.2) | |
| college | 17(29.8) | 9(16.1) | 26(23.0) | |
| university | 3(5.3) | 3(5.4) | 6(5.3) | |
| total | 57(100) | 56(100) | 113(100) | |
| **Occupation** | | | | |
| Salaried employment | 36(63.2) | 31(55.4) | 67(59.3) | 0.377[b] |
| Self-employment | 10(17.5) | 16(28.6) | 26(23.0) | |
| other | 11(19.3) | 9(16.1) | 20(17.7) | |
| total | 57(100) | 56(100) | 113(100) | |
| **No. of Children** | | | | |
| 1 | 5(8.8) | 3(5.4) | 8(7.1) | 0.478[b] |
| 2 | 22(38.6) | 16(28.6) | 38(33.6) | |
| 3 | 15(26.3) | 21(37.5) | 36(31.9) | |
| 4 | 15(26.3) | 16(28.6) | 31(27.4) | |
| total | 57(100) | 56(100) | 113(100) | |
| **Household characteristics** | | | | |
| **Gender of Household Head** | | | | |
| Male | 37(64.9) | 29(51.8) | 66(58.4) | 0.184[b] |
| female | 20(35.1) | 27(48.2) | 47(41.6) | |
| Total | 57(100) | 56(100) | 113(100) | |
| **Wealth quintile** | | | | |
| Poorest | 13(22.8) | 15(26.8) | | 0.685[b] |
| Poor | 12(21.1) | 12(21.4) | | |
| Middle | 24(42.1) | 25(44.6) | | |
| Least poor | 8(14.0) | 4(7.1) | | |
| **Mean Wealth Index score** | 0.12 | -0.12 | | 0.203[a] |
| **Child characteristics** | | | | |
| Gender Boys n (%) | 31 (54.4) | 28 (50) | | 0.708[b] |
| Girls n (%) | 26 (45.6) | 28 (50) | | |
| Mean age +SD | 106.98±25.8 | 108.89±22.5 | | 0.676[a] |
| Median (Range) | 120 (62–141) | 121 (61–140) | | 0.778[c] |

*Comparison of morbidity prevalence between the study groups at baseline.* As shown in Table 5, 59.6% and 58.4% of children from the intervention and control study groups respectively had fallen ill within the two weeks preceding the study. There were no significant differences in the prevalence [$\chi^2 = 0.219$; $p = 0.640$] and frequency [$\chi^2 = 2.648$; $p = 0.056$] of illness between the two study groups at baseline (Table 5).

**Effect of the intervention on the study outcomes.** This study was designed to test the impact of consumption of *Moringa oleifera* fortified finger millet porridge consumption on the protein and vitamin A status of children with cerebral palsy. The impact of the intervention was determined at three months after commencement of the intervention.

**Table 2. Mean nutrient dietary intakes at baseline by study groups.**

| Nutrient | units | RDA | Mean intakes ± SD | | Statistic (independent t-test) | | |
|---|---|---|---|---|---|---|---|
| | | | Intervention (N = 57) | Control (N = 56) | t-value | df | p-value |
| Energy | Kcal/d | 2000 | 1329.6±522.15 | 1324.7±574.9 | 0.047 | 111 | 0.963 |
| Proteins | g/d | 60.1 | 30.68±14.77 | 30.50±16.93 | 0.052 | 111 | 0.958 |
| Calcium | mg/d | 1300 | 844.69±199.7 | 772.31±155.2 | -1.18 | 111 | 0.241 |
| Iron | mg/d | 10 | 10.33±5.56 | 9.29±5.18 | 0.264 | 111 | 0.793 |
| Zinc | mg/d | 10 | 4.54±2.84 | 4.20±2.67 | 0.907 | 111 | 0.367 |
| Vitamin A | µg/d | 700 | 456.12±374.14 | 477.45±373.5 | 0.495 | 111 | 0.622 |
| Folate | µg/d | 300 | 200.48±103.34 | 194.72±95.4 | 0.668 | 111 | 0.506 |
| Vitamin B12 | µg/d | 1.8 | 0.64±0.49 | 0.72±0.40 | 0.411 | 111 | 0.682 |
| Vitamin C | mg/d | 45 | 73.04±41.2 | 75.00±46.36 | 1.26 | 111 | 0.310 |

SD = Standard Deviation.

**Table 3. Comparison of levels of serum albumin and retinol for study children at baseline by study groups.**

| Indicator | Mean serum levels ± SD | | Statistic (Independence t-test) | | |
|---|---|---|---|---|---|
| | Intervention Group (N = 57) | Control Group (N = 56) | t-value | df | p-value |
| Baseline Serum albumin (g/dL) | 3.067 ± 1.09 | 3.186 ± 1.16 | -1.088 | 111 | 0.280 |
| Baseline Serum retinol (µmol/L) | 0.471± 0.05 | 0.468 ± 0.02 | 0.043 | 111 | 0.966 |

**Table 4. Comparison of baseline BMI-for-age Z- scores of the study children by study groups.**

| BMI for age category | BMI Z-scores | Frequency n (%) | | Statistic | |
|---|---|---|---|---|---|
| | | Intervention N = 57 | Control N = 56 | $\chi^2$; | p-value |
| Moderately underwieght | ≥-3 to <-2 SD | 22(38.6) | 19(33.9) | 1.952; | 0.856 |
| Normal | ≥-2 to +1 SD | 30(52.6) | 31(55.4) | | |
| Overweight | >+1 to +2 SD | 3(5.3) | 4(7.1) | | |
| Obese | >+2 SD | 2(5.5) | 2(3.6) | | |

**Table 5. Comparison of morbidity prevalence of the study children at baseline.**

| Characteristic | Description | Intervention, N = 57 | | Control, N = 56 | | $\chi^2$ | p-value |
|---|---|---|---|---|---|---|---|
| | | n | % | n | % | | |
| Child sick in the last 2 weeks | | 40 | 70.2 | 37 | 66.1 | 0.219 | 0.640 |
| Nature of illness | Diarrhea | 14 | 24.6 | 11 | 19.6 | 0.397 | 0.529 |
| | Fever | 10 | 17.5 | 14 | 25.0 | 0.939 | 0.333 |
| | Cough/cold | 16 | 28.1 | 13 | 23.2 | 0.349 | 0.555 |
| | Epileptic fever | 24 | 42.1 | 17 | 30.4 | 1.686 | 0.194 |
| Frequency of illness | No sickness | 18 | 31.6 | 21 | 37.5 | 2.648 | 0.056 |
| | weekly | 20 | 35.1 | 12 | 21.4 | | |
| | monthly | 10 | 17.5 | 15 | 26.8 | | |
| | quarterly | 9 | 15.8 | 8 | 14.3 | | |

**Table 6. Average intake of porridge by the study groups during the intervention phase.**

| Description | Mean daily average (N = 60 days) | | Statistic (independent t-test) | | |
| --- | --- | --- | --- | --- | --- |
| | Intervention | Control | t-value | df | p-value |
| Amount consumed/ml | 248.07+3.81 | 248.46+2.33 | -0.682 | 118 | 0.497 |
| Amount Left over/ml | 1.93+3.81 | 1.53+2.33 | 0.682 | 118 | 0.497 |

**The average intake of porridge by the study groups during the intervention phase.** On average, 248.07±3.81 ml and 248.46±2.33 ml of the porridge servings were consumed daily by each child in the intervention and control group respectively, throughout the period of intervention. No significant differences were recorded in the daily average consumption in the porridge servings between the study groups (t [118] = -0.682; p = 0.497). Furthermore, the amount of leftovers was not significantly different between the two study groups (p>0.05) (Table 6).

**Effect of the consumption of *M. oleifera* fortified porridge on the nutrient intakes by study groups at endline.** The Nutrisurvey analysis conducted using the 24-hour dietary recall data excluded the treatments (both the experimental and placebo porridge). The intake of proteins, was 31.37±21.2 g/d and 32.6±17.4 g/d while that of vitamin A was 467.12 ±382.7 µg/d and 470.70±312.7 µg/d among the intervention and control study groups respectively. There were no significant differences between the two study groups with respect to both the amounts of proteins [t (111) = -0.087; p = 0.931] and vitamin A [t (111) = -0.04; p = 0.938], consumed (Table 7).

**Effect of the consumption of *M. oleifera* fortified porridge on the serum levels of albumin and retinol study groups at endline.** The mean levels of serum albumin for children in the intervention and control group at endline were 3.523±1.27g/dL and 3.198±1.186 g/dL respectively while the mean levels of serum retinol for intervention and control groups at endline were 0.714±0.11 µmol/L and 0.471±0.03 µmol/L respectively (Table 8). In order to establish the effect of the consumption of the *moringa*-fortified finger millet porridge on the serum levels of albumin and retinol, the two study groups were compared at endline. There were significant differences in both the mean levels of serum albumin [t (111) = 3.454; p = 0.001] and serum retinol [t (111) = 5.553; p<0.001] between the control and intervention group children at endline (Table 8).

**Table 7. Effect of consumption of fortified porridge on mean daily nutrient intakes.**

| Nutrient | Units | RDA | Mean intakes ± SD | | Statistic | | |
| --- | --- | --- | --- | --- | --- | --- | --- |
| | | | Intervention (N = 57) | Control (N = 56) | t-value | df | p-value |
| Energy | Kcal/d | 2000 | 1407.59±529.6 | 1440.7±468.6 | -0.351 | 111 | 0.726 |
| Proteins | g/d | 60.1 | 31.37±21.2 | 32.6±17.4 | -0.087 | 111 | 0.931 |
| Calcium | mg/d | 1300 | 831.63±401.6 | 811.25±346.1 | -1.03 | 111 | 0.635 |
| Iron | mg/d | 10 | 10.77±3.1 | 9.38±3.1 | 0.347 | 111 | 0.759 |
| Zinc | mg/d | 10 | 4.61±2.6 | 4.70±2.1 | -0.752 | 111 | 0.454 |
| Vitamin A | µg/d | 700 | 467.12±382.7 | 470.70±312.7 | -0.04 | 111 | 0.938 |
| Folate | µg/d | 300 | 206.60±100.3 | 196.80±83.4 | 0.803 | 111 | 0.424 |
| Vitamin B12 | µg/d | 1.8 | 0.61±0.1 | 0.65±0.3 | 0.05 | 111 | 0.961 |
| Vitamin C | mg/d | 45 | 75.6±50.4 | 75.69±40.3 | 0.625 | 111 | 0.833 |

*Statistically significant differences between the two study groups at p<0.05.

**Table 8. Effects of consumption of *M. oleifera* fortified porridge on the levels of serum albumin and retinol of the children by study groups.**

| Indicator | Mean Serum levels at | | Statistics (t-test) | | |
|---|---|---|---|---|---|
| | Control (N = 56) | Intervention (N = 57) | t | df | p-value |
| Serum albumin g/dL | 3.198±1.18 | 3.523±1.27 | 3.454 | 111 | 0.001* |
| Serum Retinol (µmol/L) | 0.471±0.03 | 0.714±0.11 | 5.553 | 111 | <0.001* |

*Significant at p<0.05.

The serum levels of albumin and retinol for each of the two study groups were compared between baseline and endline and the magnitude of change was computed (Table 9). The results showed that in the intervention group, serum albumin levels significantly increased by 0.456±0.12 g/dL [t (56) = -3.890; p <0.001] while serum retinol levels significantly increased by 0.243±0.10 µmol/L [t (56) = -6.747; p <0.001. On the other hand, among children in the control group, the levels of serum albumin and retinol only increased marginally by 0.012 ±0.07 g/dL [t (55) = -0.167; p = 0.868] and 0.0021±0.02 µmol/L [t (55) = -0.139; p = 0.890] respectively (Table 9).

The differences in the magnitude of change in the serum (albumin and retinol between the control and intervention groups were further determined by computing the difference in difference (DiD). The results showed significant differences in the magnitude of change in the levels of serum albumin and serum retinol between the two study groups with the greater magnitudes of change being realized among the intervention group (Table 10).

**Effect of the consumption of *M. oleifera* fortified porridge on the weight status of the study children at endline.** The rates of undernutrition among the intervention and control group children were 10.52% and 34% respectively while 71.9% and 57.1% of children from intervention and control groups respectively, were in the normal weight range. Furthermore, the weight distribution among the children in the two study groups was significantly different at endline [χ2 = 30.985; p = 0.037] (Table 11).

**Effect of the consumption of *M. oleifera* fortified porridge on morbidity prevalence of the study children at endline.** The two groups significantly differed in both prevalence (p = 0.003) and frequency (p = 0.012) of illness. Significant differences existed concerning diarrheal morbidity (p = 0.044), coughs and cold (p = 0.041), epileptic fits (p = 0.025), and frequency of illness (p = 0.012) between the two study groups. The children in the intervention group had significantly reduced morbidity compared to the control group (Table 12).

**Table 9. Differences in baseline and endline mean levels of serum albumin and serum retinol by study groups.**

| Indicator | Group | Mean Serum levels ±SE | | MoC | Statistics (paired t-test) | | |
|---|---|---|---|---|---|---|---|
| | | Baseline | Endline | Difference | t | df | p-value |
| Serum Albumin (g/dL) | 1 | 3.067±1.09 | 3.523±1.27 | 0.456±0.12 | -3.890 | 56 | <**0.001*** |
| | 2 | 3.186±1.16 | 3.198±1.18 | 0.012±0.07 | -0.167 | 55 | 0.868 |
| Serum Retinol (µmol/L) | 1 | 0.471±0.05 | 0.714±0.11 | 0.243±0.10 | -6.747 | 56 | <**0.001*** |
| | 2 | 0.468±0.02 | 0.471±0.03 | 0.0021±0.02 | -0.139 | 55 | 0.890 |

MoC = Magnitude of Change SE = Standard Error of Mean

* Significant at p<0.05

1- Intervention group 2- Control group

**Table 10. Magnitude of change (Baseline versus Endline) on serum albumin and serum retinol levels by study groups (difference in difference).**

| Variable | Intervention N = 57 | Control N = 56 | t-value | df | t-test |
|---|---|---|---|---|---|
| Serum albumin | 0.4561 ± 0.12 | 0.0125 ±0.07 | 3.178 | 111 | 0.002* |
| Serum retinol | 0.2433 ± 0.10 | 0.0021 ±0.02 | 2.57 | 111 | <0.012* |

* Significant at p<0.05.

**Table 11. BMI-for age Z-scores of the study children at endline.**

| Description | BMI Z-scores | intervention N = 57 | control N = 56 | χ2; | p-value* |
|---|---|---|---|---|---|
| **Endline** Severely undernourished | <-3 SD | 0(0) | 2(3.6) | 30.985; | 0.037* |
| Moderately undernourished | ≥-3 to <-2 SD | 6(10.52) | 17(30.4) | | |
| Normal | ≥-2 to +1 SD | 41(71.9) | 32(57.1) | | |
| Overweight | >+1 to +2 SD | 6 (10.5) | 5(8.9) | | |
| Obese | >+2 SD | 4(7.0) | 0(0) | | |

* Significant at p<0.05.

**Table 12. Morbidity prevalence of study children at endline by study groups.**

| Morbidity in the previous 2 weeks | Intervention (N = 57) | Control (N = 56) | p-value |
|---|---|---|---|
| **Endline** | n(%) | N(%) | |
| Children who fell sick | 14(24.6) | 29(51.8) | **0.003***  |
| Children who had diarrhea | 6(10.5) | 42(75.0) | **0.044***  |
| Children who had fever | 5(8.8) | 10(17.9) | 0.155 |
| Children who had coughs/cold | 6(10.5) | 13(23.2) | **0.041***  |
| Children who had epileptic fits | 5(9.0) | 14(25.0) | **0.025***  |
| Frequency of Sickness No sickness | 44(77.2) | 28(50.0) | **0.012***  |
| 1–2 times | 4(7.0) | 7(12.5) | |
| 3–6 times | 8(14.0) | 13(23.2) | |
| ≥ 7 times | 1(1.8) | 8(14.3) | |

* Significant at p<0.05, $x^2$ test.

## Discussion

This Randomized Controlled Clinical Trial (RCT) study was undertaken to investigate the effect of the consumption of *M. oleifera* fortified porridge on protein and vitamin A status in children with cerebral palsy. Randomization was successfully conducted because the intervention and control groups were similar in their baseline characteristics.

### Nutrient intakes of the study children at baseline

The intakes of proteins and vitamin A between the two study groups showed no significant differences at baseline. At endline significant differences were observed. A Ugandan study [32]

reported that the average daily nutrient intake from other foods excluding the treatments (unfortified finger millet porridge and finger millet porridge fortified with 7% *M. oleifera* leaf powder or 17% C maxima flesh) consumed by children did not indicate a significant difference within the groups as the sampled children had been drawn from similar socio-economic characteristics. In the current study, the significant differences observed between the groups at endline concerning the serum albumin and retinol levels as well as BMI-for-age Z-scores can therefore be attributed to the differences in the nutrient content of the porridge given to each of the groups as opposed to the portion size consumed. The Ugandan study also observed that children fed on traditional millet porridges had challenges in meeting their vitamin A requirement. Feeding of nutrient-enriched foods to children with cerebral palsy is therefore important in order to make up for the usually poor food intake, and feeding challenges which are characteristic to these children. Maximizing the dietary intake of specific nutrients such as vitamin A and protein is therefore critical to health.

### Effect of *M. oleifera* fortified porridge consumption on the levels of serum albumin and serum retinol of the study children

This study was designed to test the effect of the consumption of *M. oleifera* fortified porridge on protein and vitamin A status of children with cerebral palsy. The results showed that the consumption of *M oleifera*-fortified porridge improved the levels of both serum albumin and serum retinol of the study children. At baseline serum albumin levels of both the study groups were below the normal range (3.5–5.0 g/dL) which indicated protein deficiency, secondary to poor dietary intakes or presence of infections [33,34]. At endline the serum albumin levels among the children in the intervention group were in the normal range albeit toward the lower limit (3.52 g/dL) representing a significant but modest improvement. The serum albumin levels among the children in the control group did not improve significantly at endline.

The baseline serum retinol levels for both study groups were below the cutoff values (0.7 μmol/L). At endline, the serum retinol levels among the children in the intervention group increased significantly to a value of 0.714 μmol/L, which was slightly above the cut-off for RDA while that of the children in control group experienced no significant change. Low serum retinol levels (below RDA cutoff values) represent vitamin A deficiency, which could be due to low dietary intake arising from feeding difficulties. Vitamin A deficiency has implications in reduced immunity and increased susceptibility to infections [35]. The significant increase in serum albumin and retinol to levels above the cutoff values, observed at the endline among the intervention group as compared to the control study group implies the efficacy of the *M. oleifera* fortified flour in raising/improving the serum levels thus enhancing the nutritional status of the child.

However, the findings of the current study, deviate from those of a Ghanaian study conducted by Boateng et al., [36] on improving blood retinol concentrations with complementary foods fortified with *M. oleifera* leaf powder. In the Ghanaian study, only marginal, insignificant rise in serum levels of vitamin A after 6-week supplementation with *Moringa* was observed. May be the 6-week intervention may have been too short to the demonstration of the effect of the treatment. Another study conducted in Burkina Faso reported that after 12 months of intervention with *Moringa oleifera* leaf powder, the proportion of children with low serum retinol concentration decreased significantly in both control and treatment study groups. However, there was no significant difference between the groups at endline thus attributing their improvement in serum retinol status to other factors other than the intervention [37]. Our study eliminated known confounding factors by random placement of the study participants in the study groups (Randomized Clinical Trial) unlike in the Burkina Faso study.

Furthermore, the randomization process in this study was successful in that the study groups were similar in their baseline characteristics. This implies that the significant differences observed between the groups at the endline with respect to the serum albumin and serum retinol levels can be attributed to the intervention. In a related publication, the authors have reported the improved nutrient content of a *Moringa*-fortified fermented finger millet porridge and its potential for mitigating malnutrition among nutritionally vulnerable populations [22].

### Effect of *M. oleifera* fortified porridge consumption on the weight status for the study children

BMI-for-age Z scores of the children in the two study groups were significantly different at endline (p = 0.037). The percentage of underweight children in the intervention study group decreased from 38.6% to 10.5% while that in control group only decreased from 33.9% to 30.4%. The percentage of overweight children decreased from 10.7% to 5.4% among the control children and from 8.8% to 5.3% among intervention group children. Overall, the weight status of children in the intervention group changed significantly (p<0.001) as opposed to that of the control group which did not change significantly (p = 0.109). The improvement observed in the weight status of children in the intervention study group was attributable to the consumption of the *Moringa*-fortified porridge by the children. The findings of the current study show the efficacy of *Moringa*-fortified finger millet porridge in improving the BMI–for-age Z scores of children with cerebral palsy.

### Effect of *M. oleifera* fortified porridge consumption on the prevalence of morbidity among the study children

Significant differences in the morbidity status between the groups were observed at endline with the children in the intervention groups showing significant improvement in morbidity status as opposed to the control group that showed no significant change. This improvement in morbidity among the intervention group is most likely attributable to the intervention since the two groups were similar in their morbidity patterns at baseline. A recent study conducted in Uganda reported that children fed on fermented *Moringa*-millet porridges had significantly fewer incidences of diarrhea (p = 0.006) and respiratory infections (p = 0.003) compared to children fed on traditional millet porridges. In the Uganda study, however, the Records from Voluntary Health Trainers showed that the health-seeking habits of all groups were not significantly different [32]. The findings of the current research consolidate the association between dietary intakes, disease, and nutrition status as established in the literature [12,38,39].

### Strengths of the study

Pilot-testing of the study which ensured the validity of the study and the reliability of data. Successful randomization ensured that the differences in the study outcomes at endline were majorly attributable to the intervention. The study tested the vitamin A and protein content of the two flours prior to utilization for making the porridges. The study made use of an easily prepared food fortificant (*Moringa oleifera* leaves) on available and commonly consumed food item (porridge made from millet flour). It also used a commonly available under-utilized food product i.e *Moringa oleifera*.

### Limitations of the study

Fortification level was at 10%. A higher ratio would otherwise have a significant effect on sensory acceptability of the porridge. To improve the fortificant level, further investigations are

necessary on strategies that would not interfere with the sensory characteristics of the porridge.

## Conclusion

We conclude that the consumption of *M. oleifera* fortified finger millet porridge was effective in improving the serum albumin and retinol status of the children with cerebral palsy as well as their weight status. This study recommends the promotion of *Moringa oleifera* leaf powder as a suitable fortificant for home prepared foods, such as porridge for children with this condition to improve their dietary intake of protein and vitamin A.

## Supporting information

**S1 Checklist. CONSORT checklist.**
(DOCX)

**S1 Fig. Pictures of intervention product.**
(DOCX)

**S1 Table. Socio-demographic characteristics of caregivers.**
(DOCX)

**S2 Table. Pilot study results.**
(DOCX)

**S1 File. Trial protocol.**
(DOC)

## Acknowledgments

The authors would like to acknowledge the important roles played by Ms. Teresiah Ndunda (Technical University of Kenya), Prudence Njiru (Kenya Industrial Research Development Institute), Norman Wachira (Kenya Forest Research Institute), John Gachoya (Kenyatta University), Felix Ondiek, Maurice Wagaki, David Mwilu, Lynette Nyawira, and Dan Macharia of the research team.

## Author Contributions

**Conceptualization:** Janet Kajuju Malla.

**Supervision:** Sophie Ochola, Irene Ogada, Ann Munyaka.

**Writing – original draft:** Janet Kajuju Malla.

**Writing – review & editing:** Sophie Ochola, Irene Ogada, Ann Munyaka.

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
