## [Decision Letter · Decision Letter 0]

15 Jul 2022

PGPH-D-22-00204

Effect of Moringa Oleifera fortified porridge consumption on Protein and Vitamin A status of children with Cerebral Palsy in Nairobi, Kenya: A Randomized Controlled Trial

Dear Dr. malla,

Thank you for submitting your manuscript to PLOS Global Public Health. After careful consideration, we feel that it has merit but does not fully meet PLOS Global Public Health’s publication criteria as it currently stands. Therefore, we invite you to submit a revised version of the manuscript that addresses the points raised during the review process.

The authors conduct a RCT to assess the effect of Moringa-fortified millet on the nutritional status (protein and vitamin A status) of children with cerebral palsy. The study found that compared to the controls, the intervention group demonstrated higher serum albumin and retinol and higher BMI z scores. The study is well written and adds an essential aspect to the prevention and management of undernutrition. However, the authors need to make clarifications before the manuscript is considered for publication.

**Major comments**

**Table 7** is confusing. Are these mean daily nutrient intakes during the 3 months of intervention? Why are they different from the values in **Table 2.** This difference begs the question of whether the serum retinol and albumin levels seen later are due to the intervention (Moringa) or the improved dietary diversity during the study. Please explain?Did the authors adjust serum retinol for inflammation (elevated C-reactive protein and/or α_1_-acid glycoprotein) given the high rates of infectious disease in this region? If not, please give reasons. Please see: https://www.ncbi.nlm.nih.gov/pmc/articles/PMC8352745/Assessing vitamin A status using serum retinol is expensive and requires more blood volume draws. The authors should discuss why they did not consider cheaper options— retinol binding protein?Although the tables in the main manuscript are clear, please consider transforming some of the tables with continuous outcomes into figures such as bar plots or boxplots with p-values. Then put these figures in a panel. This will reduce the number of tables.**S2 Table** (sociodemographic Table) should be included in the main manuscript as **Table 1**.Current **Figure 3** should be placed in the supplementary document.

**Minor comments**

Please change “End line” to “endline”Page 11, line 186: change “week day” to “weekday”Page 12, line 209: Change “while” to “but”Page 12, line 223: Change “It” to lower case “it.”There is a typo in this sentence “Data were tested for normality: parametric tests including test” please correct this.Please change to change “Mc Nemar’s” to “McNemar’s Test” throughoutPlease do not capitalize M in Morbidity in the caption of Table 5Page 16, line 298: Please do not capitalize “Cerebral Palsy”SD (SE) of baseline and endline serum albumin in table 9 disagree with that of table 2 and 8, respectively. Please clarify.Please add a comma instead of a period in this sentence “The rates of undernutrition among the intervention and control group children were 10.52% and 34% respectively while 71.9% and 57.1% of children from intervention and control groups respectively. Were in the normal weight range”Table 11 change “BMI category” to “BMI Z scores”When describing the difference between the interventions and control group at the end of the trial, please explicitly state that these observations are post-study (endline). This information should be in the table title.As clearly demonstrated in the study design in **Figure 2,** explicitly add the time points of 24-h recall assessment into the “Dietary intake” section.

We look forward to receiving your revised manuscript.

Kind regards,

Paddy Ssentongo, MD, PhD, MPH

Academic Editor

Journal Requirements:

1. Thank you for submitting your clinical trial to PLOS Global Public Health and for providing the name of the registry and the registration number. The information in the registry entry suggests that your trial was registered after patient recruitment began. PLOS ONE strongly encourages authors to register all trials before recruiting the first participant in a study.

1) your reasons for your delay in registering this study (after enrolment of participants started);

2) confirmation that all related trials are registered by stating: “The authors confirm that all ongoing and related trials for this drug/intervention are registered”.

3. Please amend your Data Availability Statement and indicate where the data may be found

4. Please provide separate figure files in .tif or .eps format.
---

## [Editor Report · Decision Letter 1]

29 Sep 2022

Effect of Moringa Oleifera fortified porridge consumption on Protein and Vitamin A status of children with Cerebral Palsy in Nairobi, Kenya: A Randomized Controlled Trial

PGPH-D-22-00204R1

Dear Ms. malla,

We are pleased to inform you that your manuscript 'Effect of Moringa Oleifera fortified porridge consumption on Protein and Vitamin A status of children with Cerebral Palsy in Nairobi, Kenya: A Randomized Controlled Trial' has been provisionally accepted for publication in PLOS Global Public Health.

Best regards,

Paddy Ssentongo, MD, PhD, MPH

Academic Editor